# GROOD: GRadient-aware Out-Of-Distribution detection

**Mostafa ElAraby**
*{elarabim}@mila.quebec*
*DIRO, Université de Montréal*
*Mila - Quebec AI Institute*

**Sabyasachi Sahoo**
*Université Laval, IID*
*Mila - Quebec AI Institute*

**Yann Pequignot**
*Institut Intelligence et Données (IID)*
*Université Laval*

**Paul Novello**
*DEEL, IRT Saint Exupéry*

**Liam Paull**
*DIRO, Université de Montréal*
*Mila - Quebec AI Institute*
*CIFAR AI Chair*

**Reviewed on OpenReview:** *https://openreview.net/forum?id=2V7itvvMVJ*

## Abstract

Out-of-distribution (OOD) detection is crucial for ensuring the reliability of deep learning models in real-world applications. Existing methods typically focus on feature representations or output-space analysis, often assuming a distribution over these spaces or leveraging gradient norms with respect to model parameters. However, these approaches struggle to distinguish near-OOD samples and often require extensive hyper-parameter tuning, limiting their practicality. In this work, we propose GRadient-aware Out-Of-Distribution detection (GROOD), a method that derives an OOD prototype from synthetic samples and computes class prototypes directly from In-distribution (ID) training data. By analyzing the gradients of a nearest-class-prototype loss function concerning an artificial OOD prototype, our approach achieves a clear separation between in-distribution and OOD samples. Experimental evaluations demonstrate that gradients computed from the OOD prototype enhance the distinction between ID and OOD data, surpassing established baselines in robustness, particularly on ImageNet-1k. These findings highlight the potential of gradient-based methods and prototype-driven approaches in advancing OOD detection within deep neural networks.

## 1 Introduction

Deep neural networks (DNNs) have demonstrated exceptional performance across domains such as computer vision, natural language processing, and robotics (Goodfellow et al., 2016; LeCun et al., 2015). Their success largely relies on the assumption that training and test data follow an independent and identically distributed (iid) pattern (Krizhevsky et al., 2012; Simonyan & Zisserman, 2015). However, this assumption often fails in real-world scenarios, where DNNs encounter out-of-distribution (OOD) inputs that deviate significantly from the training distribution (Hendrycks & Gimpel, 2016). As a result, models that perform well on in-distribution (ID) data frequently produce overly confident yet incorrect predictions on OOD samples, posing significant risks to safety-critical applications such as healthcare and autonomous driving (Litjens et al., 2017;

Bojarski et al., 2016). In such scenarios, it becomes imperative for the model to exhibit self-awareness about its own limitations (Gal & Ghahramani, 2015). Conventional approaches that focus solely on minimizing the training loss are often ill-equipped to cope with OOD samples, thereby jeopardizing the safe and reliable deployment of deep learning systems (Duchi & Namkoong, 2018; Arjovsky et al., 2019; Shen et al., 2019; Liu et al., 2021a).

Consequently, several active lines of research work toward equipping DNNs with the capability to effectively detect unknown or OOD samples. Among these, post-hoc OOD detection methods stand out as the most convenient, as they utilize the representations of a pre-trained DNNs, require no additional training, and can be applied to any neural network. Experimental studies using large benchmarks have underscored the effectiveness of post-hoc OOD detection methods, which has led to the development of several specialized OOD libraries (Yang et al., 2022; Zhang et al., 2023a; Kirchheim et al., 2022; Novello et al., 2023).

Existing OOD detection methods typically focus on either feature-space-based measures, which assess the distance of inputs to learned feature representations (Sun et al., 2022; Lee et al., 2018), or gradient information (Lee et al., 2022; Sun et al., 2021a; Lee & AlRegib, 2020; Chen et al., 2023), which analyze the model's gradient space. However, these methods often struggle in scenarios where OOD samples lie near class boundaries or exhibit characteristics similar to hard in-distribution ID examples. Furthermore, current approaches rarely exploit the inherent geometry of learned feature manifolds, limiting their robustness and generalization.

To address these challenges, we propose a novel post-hoc method called GRadient-aware Out-Of-Distribution detection (GROOD) that relies on feature and gradient spaces for improved OOD discrimination addressing three persistent challenges. First, it improves the detection of near-OOD samples those that are semantically close to the in-distribution by leveraging gradient vectors with respect to an artificial OOD prototype, which provide a more discriminative signal than feature or logit-based scores. Second, GROOD generalizes robustly across diverse architectures, including ResNets and Vision Transformers, where many existing methods suffer performance degradation. Third, it significantly reduces hyper-parameter sensitivity and exhibits stable AUROC performance across training epochs, alleviating the common issue of checkpoint instability. These properties make GROOD a practical and reliable post-hoc framework for real-world deployment.

Our method is inspired by two complementary observations. First, the neural collapse (NC) property (Papyan et al., 2020) suggests that, at the end of neural network training, within-class variability tends to zero for sample representations in the feature space. This motivates the use of nearest class prototype (NCP) classification, which relies on distances to class prototypes, defined as the means of the samples of each class in the feature space. To further enhance the discriminative power of this approach for out-of-distribution detection, we extend the logits to also incorporate distances to an additional OOD prototype, alongside the distances to the class prototypes (§ 4.1).

Second, we observe that OOD samples tend to exhibit a more dispersed distribution in the feature space compared to ID samples. Capitalizing on this characteristic, we introduce an artificial OOD prototype, strategically positioned to be distinct from the ID class prototypes. By then examining how sample representations respond to this OOD prototype, specifically by analyzing gradients of the NCP loss with respect to it, we can gain a more nuanced understanding of the differences between ID and OOD samples, leading to more effective discrimination in the gradient space.

Our approach differs from traditional post-hoc methods by computing gradients with respect to an artificial OOD prototype rather than the network's parameters. The magnitude of these gradients serves as a key indicator: for ID data, the OOD prototype has a relatively small influence on the confidence of the prediction in the feature space, reflecting stable classification. In contrast, for OOD data, the OOD prototype has a more substantial influence on the confidence of the prediction; meaning that a smaller shift in the OOD prototype's representation is sufficient to cause a larger change in the classification confidence. We conduct an extensive empirical study following the recent methodology introduced in the OpenOOD Benchmark (Zhang et al., 2023a), but we also evaluate our method on other recent architectures.

Our key results and contributions are summarized as follows.

- We propose GROOD, a gradient-aware OOD detection framework that integrates neural collapse geometry, gradient-space analysis, and synthetic OOD generation for robust OOD discrimination.

- We demonstrate, via an oracle experiment, that an idealized OOD prototype significantly improves OOD detection.

- We introduce a novel mixup-based approach for generating synthetic OOD data, enhancing ID/OOD decision boundaries and reducing the need for additional auxiliary OOD data.

- We conduct extensive empirical evaluations and ablations, demonstrating GROOD's effectiveness and providing new insights into the interplay of feature and gradient spaces for OOD detection.

While NC is often seen as a limitation for OOD detection, as tightly clustered ID features can cause overconfident misclassifications, GROOD turns this into a strength. Exploiting the geometric regularity of class prototypes, it detects OOD samples not via raw confidence but through their abnormal gradient sensitivity to a fixed synthetic OOD prototype. This prototype does not aim to represent the full diversity of OOD data, but serves as a consistent reference point, inducing discriminative gradient responses that separate ID and OOD samples while avoiding the typical overconfidence failure mode.

## 2 Related Work

**Neural Network Properties**   Prior research has emphasized the significance of linear interpolation within manifold spaces, with applications ranging from word embeddings (Mikolov et al., 2013) to machine translation (Hassan et al., 2017). Extending these concepts, Verma et al. (2019) proposed manifold mixup, a method that smooths decision boundaries and reduces overconfidence near ID data. Our work primarily leverages the NC property (Papyan et al., 2020; Kothapalli et al., 2022) and prototype/centroid-based classification. Specifically, we exploit the sensitivity of the OOD prototype to enhance the distinction between ID and OOD samples. To construct this OOD prototype, we employ manifold mixup to generate a synthetic OOD dataset, enabling a more robust and structured detection framework.

**History of OOD Detection**   The study of handling OOD samples has a long history, dating back to early works on classification with rejection (Chow, 1970; Fumera & Roli, 2002). These early methods introduced the idea of abstaining from classification when confidence was low, often using simple model families such as SVM (Cortes & Vapnik, 1995). The phenomenon of neural networks producing overconfident predictions on OOD data was first revealed by Nguyen et al. (2015), highlighting the need for robust detection mechanisms in modern deep learning systems. Building on this foundational work, subsequent research has focused on various techniques for detecting OOD samples, which can be broadly categorized as output-based, feature-based, and gradient-based methods.

**Output-Based Methods**   Many OOD detection approaches directly utilize the model's outputs. Maximum softmax probability, often scaled for calibration, is a classic OOD detection metric (Hendrycks & Gimpel, 2016; Guo et al., 2017). Building on this, temperature scaling combined with input perturbations has shown promise in refining the separation between ID and OOD data (Liang et al., 2018). Additionally, logits themselves have been used for OOD detection, with some methods applying metrics such as KL divergence (Hendrycks et al., 2022). Beyond these, energy-based methods compute OOD scores using energy derived from logits (Liu et al., 2020). Refinements such as truncating activations (Sun et al., 2021b; Sun & Li, 2022) or removing dominant singular values (Song et al., 2022; Djurisic et al., 2022) have been proposed to reduce overconfidence. Generalized entropy scores over logits have also emerged as a robust alternative (Liu et al., 2023). Unlike the aforementioned techniques, GROOD achieves robust OOD detection, even for samples near ID boundaries, by combining gradient norms with class prototype-based representations.

**Feature-Based Methods**   The feature space of neural networks has been a rich avenue for OOD detection. Techniques such as Mahalanobis distance from class centroids (Lee et al., 2018; Ren et al., 2021) and Gram matrices of features (Sastry & Oore, 2020) are prominent examples. Additional methods utilize noise prototypes (Huang et al., 2021a), virtual logits (Wang et al., 2022), and nearest neighbor distances (Sun

et al., 2022). Modern Hopfield networks (Zhang et al., 2023b) have also been explored for this purpose. Cosine similarity between test samples and class features (Techapanurak et al., 2020; Chen et al., 2020) has gained traction, with some methods proposing the use of singular vectors for enhanced detection (Zaeemzadeh et al., 2021). Our approach extends these ideas by incorporating an artificial OOD prototype into the feature space, creating a novel gradient-based perspective for OOD detection.

**Gradient-Based Methods** Gradient-based methods have gained attention for their ability to capture additional information beyond intermediate layers or network outputs. The seminal ODIN approach introduced input perturbations guided by gradients to enhance OOD separation (Hsu et al., 2020). Subsequent works explored the use of gradients with respect to network weights to quantify uncertainties (Lee & AlRegib, 2020; Igoe et al., 2022; Huang et al., 2021b). Another direction utilizes Mahalanobis distances between input gradients, combined with self-supervised classifiers, to detect OOD samples (Sun et al., 2021a). GradNorm calculates an OOD score based on the gradient space of the final layer weights (Huang et al., 2021b). Recent methods, like GAIA (Chen et al., 2023), leverage gradient-based attribution abnormalities with respect to the feature space, combining channel-wise features and zero-deflation patterns. In contrast, GROOD uniquely focuses on gradients with respect to an artificial OOD prototype, capturing subtle differences between ID and OOD data. This approach enables GROOD to improve detection performance by leveraging gradient information in conjunction with prototype-based representations.

## 3 Preliminaries and Notation

We first introduce the foundational problem of OOD detection and establish the notations.

### 3.1 Context and Notations

Robust deployment of machine learning models in dynamic real-world environments often requires distinguishing between in-distribution (ID) and out-of-distribution (OOD) data to ensure reliability and safety. To formalize this challenge, we consider a supervised classification problem. Let $X$ denote the input space and $Y = \{1, 2, \ldots, C\}$ the label space, where each input-output pair $(x, y)$ is sampled from a joint data distribution $P_{XY}$. The training set $\mathcal{D}_{\text{in}} = \{(x_i, y_i)\}_{i=1}^n$ is assumed to be drawn iid from $P_{XY}$. Let $P_X$ represent the marginal distribution over $X$. The marginal distribution of the in-distribution data, denoted as $P_{\text{in}}$, is assumed to be sampled from $P_X$.

The neural network $f : X \to \mathbb{R}^{|Y|}$ is trained on samples from $P_{XY}$ to produce a logit vector, subsequently used for label prediction. The architecture of $f$ is decomposed as:

$$f = f^{\text{clf}} \circ f^{\text{pen}}, \quad \text{where} \quad f^{\text{pen}} = f^{\text{mid}} \circ f^{\text{early}}. \tag{1}$$

Here, $f^{\text{early}}$ extracts low-level features, $f^{\text{mid}}$ processes mid-level representations, and $f^{\text{pen}}$ produces penultimate features. The final classification module $f^{\text{clf}}$ outputs the predictions.

### 3.2 Problem Setting: Out-of-Distribution Detection

When deploying a machine learning model in practice, the classifier should not only be accurate on ID samples but should also identify any OOD inputs as "unknown".

Formally, OOD detection can be viewed as a binary classification task. During testing, the task is to determine whether a sample $x \in X$ comes from $P_{\text{in}}$ (ID) or not (OOD). The decision can be framed as a level set estimation:

$$G_\tau(x) = \begin{cases} \text{ID}, & \text{if } S(x) \leq \tau, \\ \text{OOD}, & \text{if } S(x) > \tau, \end{cases}$$

where $S : X \to \mathbb{R}$ is a score function quantifying the likelihood of a sample belonging to the ID distribution, and $\tau$ is a threshold ensuring that a high fraction (e.g., 95%) of ID data is correctly classified.

## 4 GROOD Methodology

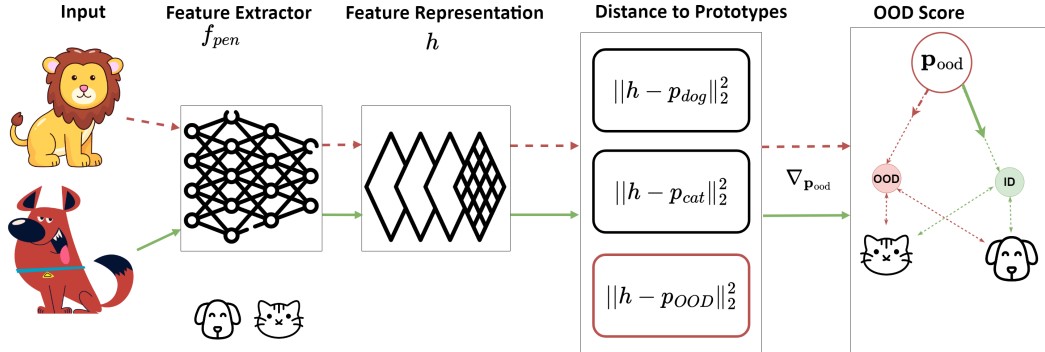

Figure 1: Initially, we build ID class prototypes as the means of the activations of ID data along with an OOD prototype capturing OOD characteristics (§ 5). Subsequently, gradients of the softmax loss built upon the NCPs distance as logits are computed w.r.t. OOD prototype (§ 4.1). Finally, the OOD score is determined using nearest neighbor distance in the gradient space (§ 4.2).

**Overview**   In this section, we introduce our proposed method GRadient-aware Out-Of-Distribution detection (GROOD), a novel framework for distinguishing between ID and OOD samples. To illustrate the core mechanism, We assume the existence of an OOD prototype (see § 5.1), and use it to compute gradients and define our OOD score.

The method comprises two primary components, illustrated in Figure 1: (1) A gradient computation framework that quantifies sample responses to an OOD prototype (§ 4.1), and (2) A nearest-neighbor scoring mechanism operating in gradient space (§ 4.2).

### 4.1 Gradients Computation

Guided by the observations in § 1, we build a distance-based classification framework that integrates both class prototypes and an artificial OOD prototype. Under the NC property (Papyan et al., 2020), ID features concentrate around their class prototypes, making distances a natural choice for logits. In contrast, the OOD prototype is positioned away from these clusters and used as a reference point, allowing us to capture the difference between ID and OOD through their gradients.

For a feature vector $h$ in the penultimate layer space, we define the logit vector as

$$L(h) = -[\|h - \mathbf{p}_1^{\text{pen}}\|_2, \ldots, \|h - \mathbf{p}_C^{\text{pen}}\|_2, \|h - \mathbf{p}_{\text{ood}}^{\text{pen}}\|_2], \tag{2}$$

where $\mathbf{p}_i^{\text{pen}}$ represents the prototype for class $i$, and $\mathbf{p}_{\text{ood}}^{\text{pen}}$ denotes the OOD prototype. These negative distances are transformed into probabilities through the softmax function:

$$p_i(h) = \frac{\exp(L_i(h))}{\sum_{j=1}^{C+1} \exp(L_j(h))}, \quad i = 1, \ldots, C+1, \tag{3}$$

where $p_{C+1}(h) = p_{\text{ood}}(h)$ represents the probability of the sample being OOD.

This formulation enables us to quantify, through the NCP loss, how well a sample aligns with the ID prototypes versus the OOD prototype. For an ID sample, we expect strong alignment with one of the class prototypes and weak alignment with the OOD prototype.

For a given feature vector $h$ and a class $y \in 1, .., C, C+1$, the cross-entropy loss associated with the NCP output $[p_i(h)]_{i=1}^{C+1}$ from equation 3 is given by:

$$H(h, y) = -\log p_y(h), \tag{4}$$

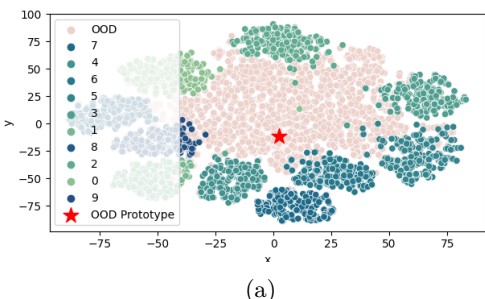 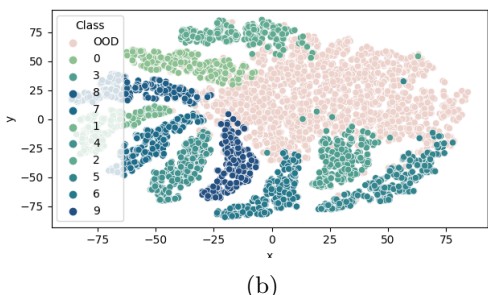

(a)          (b)

Figure 2: t-SNE plots on CIFAR-10 with ResNet-18. (a) Feature space: ID clusters vs. dispersed OOD. (b) Gradient space: clearer ID/OOD separation.

The key insight of our method lies in analyzing the gradient of the loss $H(h, y)$ for some (any) iid class $y$ with respect to the OOD prototype $\mathbf{p}_{\text{ood}}^{\text{pen}}$. Intuitively, this quantity represents the update vector for the OOD prototype assuming that the feature $h$ corresponds to an iid sample. As derived in appendix A.6, this gradient can be expressed in closed form as follows:

$$\nabla H(h) := \nabla_{\mathbf{p}_{\text{ood}}^{\text{pen}}} H(h, C + 1) = p_{\text{ood}}(h) \frac{h - \mathbf{p}_{\text{ood}}^{\text{pen}}}{\|h - \mathbf{p}_{\text{ood}}^{\text{pen}}\|_2}. \tag{5}$$

These computations are grounded in two core observations. The first is inspired by the Neural Collapse phenomenon (Papyan et al., 2020), which shows that well-trained networks often align penultimate-layer features with their corresponding class prototypes. The second is that OOD samples tend to lie outside these tight clusters, often in low-density or dispersed regions. To leverage these properties, we adopt the $\ell_2$ distance as our metric. This choice is not arbitrary: it ensures that the gradient simplifies to the closed-form in equation 5, where the gradient's norm directly equals the OOD softmax probability $p_{\text{ood}}(h)$, and its direction $(h - \mathbf{p}_{\text{ood}}^{\text{pen}})/\|h - \mathbf{p}_{\text{ood}}^{\text{pen}}\|_2$ encodes the feature's geometric deviation from the prototype. Such a direct coupling between magnitude, direction, and OOD likelihood is essential to GROOD's mechanism and would not arise with alternative metrics like cosine similarity.

For ID samples, we observe smaller gradient norms due to lower OOD probabilities. However, the complete gradient vector provides richer information than the norm alone, encoding both magnitude and directional differences between ID and OOD samples. This allows us to detect OOD samples through both the size of the hypothetical update to $\mathbf{p}_{\text{ood}}^{\text{pen}}$ and its direction.

To analyze the potential for enhanced separability, fig. 2 visualizes (a) the penultimate feature space and (b) the gradient space using t-SNE. Our intuition is that the gradient $\nabla H(h)$ will yield a more distinct separation between ID and OOD samples compared to the feature space. As seen in fig. 2 (a), ID samples form class-specific clusters, while OOD samples are scatttered in the space. However, the t-SNE plot of the gradient space in fig. 2 (b) reveals a different spatial arrangement, suggesting that the gradient transformation highlights discriminative characteristics for OOD detection, which GROOD leverages through distance computations in this space (eq. (6)).

## 4.2 Final OOD Score Computation

Having established how to compute discriminative gradients with respect to an OOD prototype, we now address a key challenge: how to effectively use these gradients to distinguish between ID and OOD samples. Our solution leverages the observation that ID samples produce similar gradient patterns, while OOD samples generate distinctly different ones.

### 4.2.1 Distance-Based Scoring

For a test sample $x_{\text{new}}$, we first compute its feature representation $h(x_{\text{new}})$ and corresponding gradient $\nabla H(h(x_{\text{new}}))$. Our OOD score is then defined as the distance to the nearest training gradient:

$$S(x_{\text{new}}) = \min_{x \in \mathcal{D}_{\text{in}}} \|\nabla H(h(x_{\text{new}})) - \nabla H(h(x))\|_2 \tag{6}$$

This formulation captures our intuition that OOD samples will produce gradients that deviate significantly from those seen during training. The minimum distance provides a natural measure of "outlierness" - the further a gradient is from its nearest training neighbor, the more likely the sample is to be OOD.

**Efficient Implementation**   A naive implementation of nearest neighbor search in high-dimensional gradient space would be computationally prohibitive. We address this challenge using the FAISS library Douze et al. (2024), which provides efficient approximate nearest neighbor search through inverted lists and quantization. This makes our method practical for large-scale applications while maintaining accuracy.

The preceding section outlined the core methodology of GROOD, detailing how gradients with respect to an OOD prototype are computed and subsequently used within a nearest-neighbor scoring mechanism to distinguish ID and OOD samples. A critical element underpinning the effectiveness of this methodology is the choice and construction of the OOD prototype, $\mathbf{p}_{\text{ood}}^{\text{pen}}$. The nature and location of this prototype in the feature space directly influence the direction and magnitude of the computed gradients, thereby impacting the separability of ID and OOD samples in the gradient space. Therefore, the following section delves into the specific strategies we employ for **Prototype Computation** (§ 5), exploring various approaches to define both the class prototypes and, crucially, the OOD prototype. These approaches are designed to yield a $\mathbf{p}_{\text{ood}}^{\text{pen}}$ that optimizes the discriminative power of the gradient-based OOD score introduced in our methodology.

## 5 Prototype Computation

### 5.1 Class and OOD Prototypes

The foundation of our method lies in computing class-discriminative prototypes. For each class, we compute prototypes at both early and penultimate layers as the average of feature vectors:

$$p_y^l = \frac{1}{|X_y|} \sum_{x \in X_y} f^l(x), \quad l \in \{\text{early}, \text{pen}\} \tag{7}$$

where $X_y$ is the set of training instances in class $y$.

Similarly, the OOD prototype is computed as the average of feature vectors from a dataset $X_{ood}$:

$$p_{ood}^{\text{pen}} = \frac{1}{|X_{ood}|} \sum_{x \in X_{ood}} f^{\text{pen}}(x) \tag{8}$$

The less variability in the representation space of each ID class as per NC(Papyan et al., 2020), the more effective GROOD will be in distinguishing ID and OOD samples.

Clearly, the choice of dataset $X_{ood}$ will have a significant impact on GROOD's ability to differentiate between ID and OOD samples. In the remainder of this section we will first show that, given access to actual OOD data, GROOD performs significantly better than the state of the art (Sec. 5.1.1). Subsequently, we propose a method for synthesizing the OOD data used to calculate the OOD prototype that approximates the performance of the priviliged case (Sec. 5.2).

### 5.1.1 "Oracle" Experiment

To validate the core idea that an OOD prototype can help distinguish ID and OOD data, we performed an "oracle" experiment, assuming temporary access to some OOD information.

Table 1: Oracle experiment results compared to SOTA excluding GROOD from Table 2. AUROC (%) on far-OOD and near-OOD detection using 100 prototype samples. Results are averaged over different checkpoints; standard deviations in parentheses.

| ID Dataset | Architecture | Local Oracle | | Global Oracle | | SOTA | |
|---|---|---|---|---|---|---|---|
| | | AUROC (%) ↑ | | | | | |
| | | Far-OOD | Near-OOD | Far-OOD | Near-OOD | Far-OOD | Near-OOD |
| CIFAR-10 | ResNet-18 | 96.7 ($\pm$0.2) | 95.4 ($\pm$0.1) | 94.8 ($\pm$0.1) | 90.8 ($\pm$0.1) | 94.7 | 90.7 |
| CIFAR-100 | ResNet-18 | 94.8 ($\pm$0.2) | 85.5 ($\pm$0.5) | 88.1 ($\pm$0.03) | 82.1 ($\pm$0.1) | 82.4 | 81.3 |
| ImageNet-200 | ResNet-18 | 98.8 ($\pm$0.2) | 92.1 ($\pm$0.2) | 94.22 ($\pm$0.02) | 84.1 ($\pm$0.2) | 93.16 | 82.9 |
| ImageNet-1K | ResNet-50 | 97.9 | 91.0 | 96.2 | 79 | 95.1 | 78.1 |

**Local Oracle (Idealized):** For each ID-OOD test pair, we used a small sample (100) from the *test* OOD data to build a specific OOD prototype. We then tested the remaining OOD data. This setup essentially asks: "If we had perfect knowledge of a small subset of the specific OOD data we'd encounter, how well could our method perform?". The remarkable performance achieved in this setting (over 95% AUROC on far-OOD detection, as shown in table 1) highlights the inherent potential of an OOD prototype tailored to the specific distributional shift.

**Global Oracle (Generalizable):** For each ID dataset, we used a small "validation" portion of *all other* OOD datasets to create a single, general OOD prototype. We then tested on the held-out portion of each specific OOD dataset. This setup aims to mimic a scenario where we have access to some diverse OOD data (the validation sets) but not the specific OOD data we are currently testing on. The results from the Global Oracle provide insights into how well a more general OOD prototype can generalize across different out-of-distribution scenarios.

Despite the strong performance in the far-OOD detection tasks under both oracle settings, the notably lower performance on near-OOD detection underscores the inherent difficulty of distinguishing between distributions that are semantically or statistically close to the in-distribution data. Nevertheless, the oracle experiments strongly suggest that the concept of an OOD prototype holds significant promise for OOD detection when a representative prototype can be effectively determined.

On the other hand, in real-world scenarios, we cannot construct this oracle prototype using test OOD data. This raises a key question: *How can we approximate this optimal OOD prototype without access to the test distribution?*

## 5.2 Practical OOD Prototype Construction

The oracle experiments presented in § 5.1 demonstrated the significant potential of employing an OOD prototype to distinguish between ID and OOD data, achieving high performance when even approximate knowledge of the OOD distribution was available. This motivates our goal: to effectively approximate such an optimal OOD prototype, $\mathbf{p}_{\text{ood}}^{\text{pen}}$, without requiring access to the specific test OOD distribution, which is unavailable in practical scenarios.

While real-world OOD samples exhibit considerable diversity, representing them with a single prototype $\mathbf{p}_{\text{ood}}^{\text{pen}}$ proves effective within our gradient-aware framework. Our approach does not aim to represent all OOD samples geometrically, but rather uses the prototype as a fixed reference point. The core OOD score relies on the sensitivity of the NCP loss to this prototype, measured via the gradient $\nabla_{\mathbf{p}_{\text{ood}}^{\text{pen}}} H(h)$ (§ 4, eq. (5)). We hypothesize that OOD samples, inherently deviating from learned ID manifolds, exhibit distinct gradient sensitivity patterns (both magnitude and direction) relative to this OOD reference point, allowing separation from more stable ID samples.

We propose several complementary approaches to construct $\mathbf{p}_{\text{ood}}^{\text{pen}}$, leveraging information from an auxiliary OOD dataset $X_{ood}$:

**Synthetic OOD Generation using mixup**   Our first approach requires no external OOD data, instead synthesizing OOD-like features by exploiting decision boundaries. We perform guided prototype interpolation towards the second-highest predicted class $c_2$ at an early layer (after the first block):

$$\hat{h}(x) = f^{\mathrm{mid}} \left( \lambda f^{\mathrm{early}}(x) + (1 - \lambda)\mathbf{p}_{c_2}^{\mathrm{early}} \right) \tag{9}$$

where $\lambda = 0.5$ positions the synthetic samples near decision boundaries. This approach leverages our observation that early layer representations are more sensitive to perturbations, making them ideal for generating OOD-like features. Although effective, mixup-generated OOD samples interpolate between ID classes and may not fully capture OOD diversity.

**Auxiliary OOD Validation**   When available, we can utilize a small auxiliary OOD validation set as $X_{ood}$ to construct the prototype following eq. (8) using 100 OOD validation samples. Importantly, we ensure these samples have no category overlap with the test set. Our method shows remarkable stability to the specific choice of validation samples, with a maximum AUROC standard deviation of only 0.5% across five different random selections.

**Proximity-Based OOD Filtering (Postprocessing)**   Initially, we explored constructing the OOD prototype by simply averaging feature vectors from all available OOD samples. However, this approach yielded prototypes that lacked sufficient discriminative power, resulting in poor separation between ID and OOD data. To address this, we introduce a proximity-based filtering step to refine the OOD prototype, enhancing its ability to distinguish OOD samples from ID samples.

Specifically, given a set of candidate OOD feature vectors, we discard OOD samples whose distance $d_i = \min_j \|f^{\mathrm{pen}}(o_i) - p_j\|_2$ falls below an adaptive threshold $\tau$, computed as the $q$-th quantile of $\{d_i\}_{i=1}^{n_{\mathrm{ood}}}$. This filtering step refines the OOD prototype by ensuring separation from ID data while preserving representativeness.

A comparison of alternative $X_{OOD}$ generation methods is presented in the appendix A.2. The results reported in the remainder of this paper utilize the OOD prototype derived from synthetic data generated from ID samples.

## 6   Experiments

For a comprehensive evaluation of GROOD's performance, we adhere to the OpenOOD v1.5 criteria Zhang et al. (2023a); Yang et al. (2022). Results are aggregated in table 2 including the performance of the nearest class prototype used instead of the classification head. For robustness, each evaluation metric except for ImageNet-1k is derived from three runs with unique initialization seeds. In the case of ImageNet-1k, we report results based on a single seed run provided by torchvision maintainers & contributors (2016).

**Experimental Setup**   We evaluate performance using the Area Under the Receiver Operating Characteristic curve (AUROC), where higher values are better. Our benchmarking strategy follows the OpenOOD framework Zhang et al. (2023a); Yang et al. (2022), involving four core ID datasets (CIFAR-10, CIFAR-100, ImageNet-200, ImageNet-1k) and examining both near and far-OOD scenarios. For CIFAR-10/100 (50k train/10k test images each), near-OOD datasets are CIFAR-100/TinyImageNet, and far-OOD are MNIST, SVHN, Textures, and Places365. For ImageNet-200 (200 classes, 64x64 resolution), near-OOD datasets are SSB-hard/NINCO, and far-OOD are iNaturalist, Textures, and OpenImage-O; ImageNet-1k shares these OOD datasets. Regarding configuration, we deploy ResNet-18 for CIFAR-10/100 and ImageNet-200 using pre-trained checkpoints from OpenOOD for consistency, testing with three distinct seeds for robustness. For ImageNet-1k, we apply pre-trained torchvision models (ResNet-50, ViT-B-16, Swin-T) to explore GROOD's effectiveness in a broader context than OpenOOD v1 Yang et al. (2022). To allow for reproducibility and facilitate further research, the complete code, including training and evaluation scripts, is available at: `https://github.com/mostafaelaraby/Gradient-Aware-OOD-Detection`.

For CIFAR-10/100 and ImageNet-200, we train ResNet-18 models for 100 epochs using SGD with momentum 0.9, weight decay 5e-4, cosine learning rate decay (starting from 0.1), and batch sizes of 128 (CIFAR) and 256

(ImageNet-200). For ImageNet-1K, we use pretrained models from `torchvision` (ResNet-50, ViT, Swin). When fine-tuning is required, we follow the OpenOOD v1.5 protocol (Zhang et al., 2023a) with 30 epochs, learning rate 0.001, and batch size 256.

Table 2: Main results from OpenOOD v1.5 on standard OOD detection (AUROC). GROOD using synthetic OOD data (§ 5.2) shows superior results compared to existing baselines.

| | CIFAR-10 | | CIFAR-100 | | ImageNet-200 | | ImageNet-1K | |
|---|---|---|---|---|---|---|---|---|
| ID Acc. (%) | $95.06\%_{(\pm 0.30)}$ | | $77.25\%_{(\pm 0.10)}$ | | $86.37\%_{(\pm 0.08)}$ | | 76.18% | |
| NCP Acc. (%) | $95.01\%_{(\pm 0.08)}$ | | $77.10\%_{(\pm 0.001)}$ | | $85.75\%_{(\pm 0.003)}$ | | 71.38% | |
| Method | Near-OOD (%) ↑ | Far-OOD (%) ↑ | Near-OOD (%) ↑ | Far-OOD (%) ↑ | Near-OOD (%) ↑ | Far-OOD (%) ↑ | Near-OOD (%) ↑ | Far-OOD (%) ↑ |
| OpenMax Bendale & Boult (2015) | $87.62\%_{(\pm 0.29)}$ | $89.62\%_{(\pm 0.19)}$ | $76.41\%_{(\pm 0.41)}$ | $79.48\%_{(\pm 0.41)}$ | $80.27\%_{(\pm 0.10)}$ | $90.20\%_{(\pm 0.17)}$ | 74.77% | 89.26% |
| MSP Hendrycks & Gimpel (2016) | $88.03\%_{(\pm 0.25)}$ | $90.73\%_{(\pm 0.43)}$ | $80.27\%_{(\pm 0.11)}$ | $77.76\%_{(\pm 0.44)}$ | $83.34\%_{(\pm 0.06)}$ | $90.13\%_{(\pm 0.09)}$ | 76.02% | 85.23% |
| ODIN Liang et al. (2018) | $82.87\%_{(\pm 1.85)}$ | $87.96\%_{(\pm 0.61)}$ | $79.90\%_{(\pm 0.11)}$ | $79.28\%_{(\pm 0.21)}$ | $80.27\%_{(\pm 0.08)}$ | $91.71\%_{(\pm 0.19)}$ | 74.75% | 89.47% |
| MDS Lee et al. (2018) | $84.20\%_{(\pm 2.40)}$ | $89.72\%_{(\pm 1.36)}$ | $58.69\%_{(\pm 0.09)}$ | $69.39\%_{(\pm 1.39)}$ | $61.93\%_{(\pm 0.51)}$ | $74.72\%_{(\pm 0.26)}$ | 55.44% | 74.25% |
| EBO Liu et al. (2020) | $87.58\%_{(\pm 0.46)}$ | $91.21\%_{(\pm 0.92)}$ | $80.91\%_{(\pm 0.08)}$ | $79.77\%_{(\pm 0.61)}$ | $82.50\%_{(\pm 0.05)}$ | $90.86\%_{(\pm 0.21)}$ | 75.89% | 89.47% |
| ReAct Sun et al. (2021b) | $87.11\%_{(\pm 0.61)}$ | $90.42\%_{(\pm 1.41)}$ | $80.77\%_{(\pm 0.05)}$ | $80.39\%_{(\pm 0.49)}$ | $81.87\%_{(\pm 0.98)}$ | $92.31\%_{(\pm 0.56)}$ | 77.38% | 93.67% |
| DICE Sun & Li (2022) | $78.34\%_{(\pm 0.79)}$ | $84.23\%_{(\pm 1.89)}$ | $79.38\%_{(\pm 0.23)}$ | $80.01\%_{(\pm 0.18)}$ | $81.78\%_{(\pm 0.14)}$ | $90.80\%_{(\pm 0.31)}$ | 73.07% | 90.95% |
| GradNorm Huang et al. (2021b) | $54.90\%_{(\pm 0.98)}$ | $57.55\%_{(\pm 3.22)}$ | $70.13\%_{(\pm 0.47)}$ | $69.14\%_{(\pm 1.05)}$ | $72.75\%_{(\pm 0.48)}$ | $84.26\%_{(\pm 0.87)}$ | 72.96% | 90.25% |
| MLS Hendrycks et al. (2022) | $87.52\%_{(\pm 0.47)}$ | $91.10\%_{(\pm 0.89)}$ | $81.05\%_{(\pm 0.07)}$ | $79.67\%_{(\pm 0.57)}$ | $82.90\%_{(\pm 0.04)}$ | $91.11\%_{(\pm 0.19)}$ | 76.46% | 89.57% |
| VIM Wang et al. (2022) | $88.68\%_{(\pm 0.28)}$ | $93.48\%_{(\pm 0.24)}$ | $74.98\%_{(\pm 0.13)}$ | $81.70\%_{(\pm 0.62)}$ | $78.68\%_{(\pm 0.24)}$ | $91.26\%_{(\pm 0.19)}$ | 72.08% | 92.68% |
| KNN Sun et al. (2022) | $90.64\%_{(\pm 0.20)}$ | $92.96\%_{(\pm 0.14)}$ | $80.18\%_{(\pm 0.15)}$ | $82.40\%_{(\pm 0.17)}$ | $81.57\%_{(\pm 0.17)}$ | $93.16\%_{(\pm 0.22)}$ | 71.10% | 90.18% |
| SHE Zhang et al. (2023b) | $81.54\%_{(\pm 0.51)}$ | $85.32\%_{(\pm 1.43)}$ | $78.95\%_{(\pm 0.18)}$ | $76.92\%_{(\pm 1.16)}$ | $80.18\%_{(\pm 0.25)}$ | $89.81\%_{(\pm 0.61)}$ | 73.78% | 90.92% |
| ASH Djurisic et al. (2022) | $75.27\%_{(\pm 1.04)}$ | $78.49\%_{(\pm 2.58)}$ | $78.20\%_{(\pm 0.15)}$ | $80.58\%_{(\pm 0.66)}$ | $82.38\%_{(\pm 0.19)}$ | $93.90\%_{(\pm 0.27)}$ | 78.17% | 95.1% |
| GAIA Chen et al. (2023) | $85.1\%_{(\pm 10.2)}$ | $92.1\%_{(\pm 2.9)}$ | $70.75\%_{(\pm 2.11)}$ | $86.2\%_{(\pm 5.1)}$ | $75.1\%_{(\pm 9.8)}$ | $88.14\%_{(\pm 1.8)}$ | 66.98% | 90.2% |
| CIDER Ming et al. (2023) | $90.7\%_{(\pm 0.1)}$ | $\mathbf{94.7\%}_{(\pm 0.36)}$ | $73.10\%_{(\pm 0.3)}$ | $80.49\%_{(\pm 0.68)}$ | $80.58\%_{(\pm 1.7)}$ | $90.66\%_{(\pm 1.6)}$ | 68.9% | 92.18% |
| GEN Liu et al. (2023) | $88.2\%_{(\pm 0.3)}$ | $91.35\%_{(\pm 0.55)}$ | $\mathbf{81.31\%}_{(\pm 0.1)}$ | $79.68\%_{(\pm 0.6)}$ | $82.9\%_{(\pm 0.34)}$ | $91.36\%_{(\pm 0.45)}$ | 76.85% | 89.76% |
| fdbd Liu & Qin (2023) | $90.4\%_{(\pm 0.12)}$ | $93.16\%_{(\pm 0.25)}$ | $81.2\%_{(\pm 0.05)}$ | $79.85\%_{(\pm 0.15)}$ | $\mathbf{84.2\%}_{(\pm 0.3)}$ | $93.4\%_{(\pm 0.2)}$ | 76.6% | 92.7% |
| NCI Liu & Qin (2025) | $88.8\%_{(\pm 0.1)}$ | $91.26\%_{(\pm 0.2)}$ | $81\%_{(\pm 0.2)}$ | $81.3\%_{(\pm 0.15)}$ | $83.5\%_{(\pm 0.4)}$ | $\mathbf{93.7\%}_{(\pm 0.15)}$ | 78.6% | $\mathbf{95.5\%}$ |
| **GROOD (Ours)** | $\mathbf{91.16\%}_{(\pm 0.001)}$ | $93.8\%_{(\pm 0.02)}$ | $78.9\%_{(\pm 0.05)}$ | $\mathbf{84.44\%}_{(\pm 0.9)}$ | $\mathbf{83.4\%}_{(\pm 0.12)}$ | $92.19\%_{(\pm 0.12)}$ | $\mathbf{78.91\%}$ | 94.8 % |

**Main Results Discussion** GROOD shows strong performance across datasets, but performance varies depending on the trade-off between Near and Far-OOD detection. On CIFAR-100, GROOD achieves state-of-the-art Far-OOD AUROC (84.44%) among post-hoc methods and competitive Near-OOD performance (78.9%). While some methods like VIM (Wang et al., 2022) (81.70%) report higher Near-OOD scores, they trade off Far-OOD robustness.

On ImageNet-1k, GROOD achieves a top Far-OOD score (94.8%) but a lower Near-OOD score (78.91%), whereas CombOOD (Rajasekaran et al., 2024) yields 95.22% Near-OOD and 90.24% Far-OOD.

These results reflect a key characteristic of GROOD: its effectiveness depends on the structure of the ID feature space. As GROOD relies on geometric separation of class prototypes (inspired by Neural Collapse (Papyan et al., 2020)), its performance can degrade when ID representations are less well-clustered. Additionally, the choice of OOD prototype impacts this trade-off. For example, using an "ID-corrupted val" prototype improves Near-OOD AUROC to 80.27% (CIFAR-100) and 83.5% (ImageNet-1k), while maintaining strong Far-OOD scores.

Importantly, many top-performing methods on the OpenOOD leaderboard require access to OOD data (e.g., OE, CIDER). GROOD remains fully post-hoc and training-free, making it more practical for deployment across varied scenarios.

Table 3: Performance comparison (AUROC %) on ImageNet-1K using different architectures.

| | ViT-B-16 | | Swin-T | |
|---|---|---|---|---|
| Method | Near-OOD (%) ↑ | Far-OOD (%) ↑ | Near-OOD (%) ↑ | Far-OOD (%) ↑ |
| **GROOD (OURS)** | **76.47%** | **90.84%** | **76.10%** | 88.90% |
| ReACT Sun et al. (2021b) | 69.26% | 85.69% | 75.64% | 88.23% |
| GradNorm Huang et al. (2021b) | 39.28% | 41.75% | 47.58% | 35.47% |
| KNN Sun et al. (2022) | 74.11% | 90.60% | 71.62% | **89.37%** |
| ASH Djurisic et al. (2022) | 53.21% | 51.56% | 46.47% | 44.64% |

**Performance on Transformer Architectures** Further demonstrating the robustness and generalization capabilities of our approach, table 3 presents the OOD detection performance on ImageNet-1K using Transformer-based architectures, ViT-B-16 (Dosovitskiy et al., 2020) and Swin-T (Liu et al., 2021b).

GROOD maintains strong performance, achieving the top AUROC scores for both Near-OOD and Far-OOD detection on ViT-B-16 (Dosovitskiy et al., 2020), and the best Near-OOD score on Swin-T (Liu et al., 2021b) while being highly competitive for Far-OOD. This contrasts significantly with several other methods, such as GradNorm and ASH, whose performance severely degrades on these Transformer architectures compared to their ResNet-based results. This suggests that GROOD's mechanism, relying on gradient sensitivity relative to class and OOD prototypes, generalizes more effectively across fundamentally different architectural paradigms than methods potentially more sensitive to specific CNN feature properties.

## 6.1 Ablation Studies

To understand the contribution of each component in GROOD, we conduct a series of ablation studies on CIFAR-10 (table 4). Each study isolates a key design choice and quantifies its impact on both near-OOD and far-OOD detection performance.

Table 4: Ablation study using different losses and OOD scores to show the effectiveness of each proposed part. Evaluation done on CIFAR-10.

| Model Variant | AUROC (%) | |
| --- | --- | --- |
| | **Far-OOD** | **Near-OOD** |
| (1) Distance to the Noise prototype | $84.3_{(\pm6.1)}$ | $79.9_{(\pm6.5)}$ |
| (2) Gradient L1-norm only | $92.4_{(\pm0.48)}$ | $89.35_{(\pm0.41)}$ |
| (3) Grads. wrt class prototypes | $92.7_{(\pm0.15)}$ | $89.9_{(\pm0.05)}$ |
| (4) OOD prototype with uniform noise only | $91.7_{(\pm0.6)}$ | $88.1_{(\pm0.55)}$ |
| **GROOD** | $\mathbf{93.8}_{(\pm0.02)}$ | $\mathbf{91.16}_{(\pm0.001)}$ |

**Distance vs. Gradient-based Scoring** Our first experiment examines whether the gradient computation is truly necessary. We compare directly using the distance to the OOD prototype against our full gradient-based approach. The significant performance gap (79.9% vs. 91.16% for near-OOD and 84.3% vs 93.8% for far-OOD) demonstrates that gradients capture richer information about sample distribution than raw distances alone.

**Nearest Neighbor vs. Gradient Norm** While prior work like GradNorm Huang et al. (2021b) uses L1-norm of gradients as the OOD score, we hypothesized that our full approach, GROOD, would be more informative. The results support this: GROOD achieves 93.8% AUROC on far-OOD detection and 91.16% on near-OOD detection compared to 92.4% and 89.35% respectively with gradient norm alone, suggesting that the combination of our design choices in GROOD provides valuable signal beyond the magnitude of the gradient alone.

**OOD vs. Class Prototypes** A natural question is whether we need a dedicated OOD prototype at all - could we achieve similar results using gradients with respect to class prototypes? The experiment shows that OOD-specific prototypes provide superior performance (93.8% vs. 92.7% on far-OOD and 91.16% vs 89.9% on near-OOD), validating our design choice to explicitly model out-of-distribution behavior.

**Impact of Noise Sources** Finally, we investigate the value of our synthetic OOD data generation for OOD prototype construction compared to simply using uniform noise. Using only uniform noise degrades performance by 2.1% on far-OOD (93.8% vs 91.7%) and 3.06% on near-OOD detection (91.16% vs 88.1%), respectively, demonstrating the benefit of our comprehensive synthetic approach to prototype construction over basic uniform noise.

These ablations collectively validate GROOD's key design choices: each component contributes meaningfully to the final performance, with the full method achieving the best results across all metrics. The consistent improvements and low standard deviations ($\leq 0.6\%$) across experiments indicate the robustness of our approach.

## 6.2 Robustness to Checkpoint Choice

Table 5: Standard Deviation of AUROC for the Last 15 Epochs on CIFAR-100

| Method | Std (Far OOD) | Std (Near OOD) |
|---|---|---|
| MDS Lee et al. (2018) | 1.05 | 0.24 |
| ODIN Liang et al. (2018) | 0.40 | 0.22 |
| GradNorm Huang et al. (2021b) | 0.62 | 0.72 |
| VIM Wang et al. (2022) | 0.95 | 0.37 |
| GROOD | **0.38** | **0.2** |

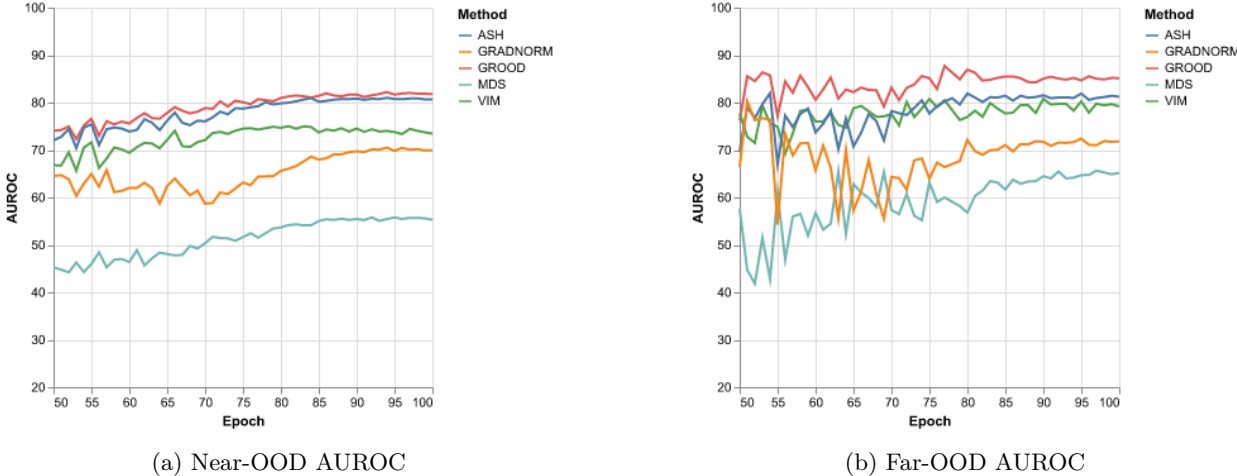

(a) Near-OOD AUROC          (b) Far-OOD AUROC

Figure 3: AUROC Performance on Cifar100 (Near and Far OOD) across different checkpoints showing the stability of GROOD

The AUROC metric, while widely used for evaluating OOD detection, can exhibit instability during training, particularly in the later stages. This instability means that small fluctuations in the model's weights can lead to significant variations in AUROC scores, making the selection of an optimal checkpoint challenging. The AUROC curve can vary sharply, even when the test error is relatively stable, indicating a sensitivity to minor weight perturbations. In contrast, GROOD's design contributes to more stable OOD detection performance. The key intuition behind GROOD's robustness lies in its focus on the sensitivity of weights relative to the OOD prototype. Throughout training, while the representation space and the OOD prototype's absolute location change as the network's weights are updated, their inherent relationship the sensitivity remains stable, leading to consistent OOD detection , as further evidenced by the reduced standard deviations shown in table 5 and fig. 3.

Further ablation experiments and detailed analysis are provided in the Appendix.

## 7 Discussion and Conclusion

In this work, we introduced GRadient-aware Out-Of-Distribution detection (GROOD), a novel approach that combines gradient information with distance metrics to improve detection of OOD samples in DNN-based image classifiers. Extensive experiments across benchmarks show that GROOD effectively detects both near and far OOD samples, performs robustly across architectures and datasets, and requires little hyper-parameter tuning, making it practical for deployment in real-world settings (Litjens et al., 2017; Bojarski et al., 2016).

GROOD is particularly well-suited to scenarios where in-distribution features form clear clusters, consistent with the Neural Collapse phenomenon (Papyan et al., 2020). In such cases, deviations from prototypes

are easily detected. However, its performance may degrade when in-distribution data is noisy or poorly separated.

From a practical perspective, GROOD improves inference speed compared to KNN-based methods (see appendix A.4), but storing and processing gradients can be costly for very large datasets or high-resolution images. Prototype construction also matters: while relatively robust, performance can depend on the diversity of auxiliary OOD samples (appendix A.2).

Finally, several design choices proved important. Proximity-based filtering of synthetic prototypes (§ 5.2) enhanced discriminative power, and mixup toward the **second**-highest predicted class yielded stronger results than random interpolation, likely because it produces harder boundary samples.

In summary, GROOD offers a simple and effective framework for OOD detection, especially when in-distribution data is well-structured. Future work could extend it to noisier domains and explore ways to further reduce computational overhead.

### Acknowledgments

This work was carried out within the DEEL project. It is supported by the DEEL Project CRDPJ 537462-18 funded by the Natural Sciences and Engineering Research Council of Canada (NSERC) and the Consortium for Research and Innovation in Aerospace in Québec (CRIAQ), together with its industrial partners Thales Canada inc, Bell Textron Canada Limited, CAE inc and Bombardier inc.[1]. The DEEL project is also part of IRT Saint Exupéry and the ANITI AI cluster[2]. The authors acknowledge the financial support from DEEL's Industrial and Academic Members and the France 2030 program – Grant agreements n°ANR-10-AIRT-01 and n°ANR-23-IACL-0002. We are grateful for the digital alliance of Canada and NVIDIA for compute infrastructure. We would also like to thank Dr. Samer Nashed for his fruitful discussions and valuable feedback during the review process.

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

# A Appendix

## A.1 GROOD Algorithm

For a comprehensive overview, the complete GROOD algorithm is outlined in algorithm 1 and algorithm 2.

---

**Algorithm 1** GROOD initialization: Compute prototypes and gradients for the training set

---

**Require:** Training set $\mathcal{D}_{\text{in}}$, trained model $f$
**Require:** mixup parameter $\lambda$
1: Compute class prototypes $\mathbf{P}_{\text{early}}$ and $\mathbf{P}_{\text{pen}}$          ▷ eq. (7)
2: Compute OOD prototype $\mathbf{p}_{\text{ood}}^{\text{pen}}$ using synthetic data generation§ 5.2      ▷ eq. (8)
3: **function** COMP_GRAD$(h, \mathbf{p}_{y=1,\cdots,C}^{\text{pen}}, \mathbf{p}_{\text{ood}}^{\text{pen}})$
4:      compute $\nabla H(h)$          ▷ eq. (5) using $\mathbf{p}_{y=1,\cdots,C}^{\text{pen}}, \mathbf{p}_{\text{ood}}^{\text{pen}}$
5:      **return** $\nabla H(h)$
6: **end function**
7: **for** each $x \in \mathcal{D}_{\text{in}}$ **do**
8:      $\nabla(x) = $ COMP_GRAD $(h(x), \mathbf{p}_{y=1,\cdots,C}^{\text{pen}}, \mathbf{p}_{\text{ood}}^{\text{pen}})$
9: **end for**
10: **return** $\{\nabla(x)\}_{x \in \mathcal{D}_{\text{in}}}$, $\mathbf{p}_{y=1,\cdots,C}^{\text{early}}$, $\mathbf{p}_{y=1,\cdots,C}^{\text{pen}}$, $\mathbf{p}_{\text{ood}}^{\text{pen}}$

---

---

**Algorithm 2** OOD score using GROOD

---

**Require:** Training dataset $\mathcal{D}_{\text{in}}$, trained model $f$
**Require:** $\{\nabla(x))\}_{x \in \mathcal{D}_{\text{in}}}$, $\mathbf{p}_{y=1,\cdots,C}^{\text{pen}}$, $\mathbf{p}_{\text{ood}}^{\text{pen}}$ from GROOD initialization
**Require:** function COMP_GRAD (in Alg. 1)
**Require:** Sample $x_{\text{new}}$, threshold $\tau$
1: $\nabla(x_{\text{new}}) = $ COMP_GRAD$(h(x_{\text{new}}), \mathbf{p}_{y=1,\cdots,C}^{\text{pen}}, \mathbf{p}_{\text{ood}}^{\text{pen}})$
2: Compute OOD score using Nearest Neighbor search: $S(x_{\text{new}}) = \min_{x \in \mathcal{D}_{\text{in}}} \|\nabla(x_{\text{new}}) - \nabla(x)\|_2$
3: **return** ID if $S(x_{\text{new}}) \leq \tau$ else OOD

---

## A.2 Choice of OOD data for OOD prototype computation

**OpenOOD Val** To form $\mathbf{p}_{\text{ood}}^{\text{pen}}$, we selected 100 data points from an auxiliary OOD validation dataset, as per the OpenOOD framework Zhang et al. (2023a); Yang et al. (2022), ensuring no category overlap with test set images. This selection criterion aligns with practices in established post-hoc analyses Lee et al. (2022; 2018); Kong & Ramanan (2021). Our method demonstrated robustness to the specific choice of OOD samples. An investigation with five distinct sets of 100 OOD validation samples each revealed negligible variation in AUROC, with a maximum standard deviation of **0.5%**, underscoring our approach's stability across different OOD selections.

Table 6: OOD Detection Performance (AUROC %) by Dataset and OOD Prototype Construction Method.

| OOD Prototype | Cifar-10 | | Cifar-100 | | ImageNet-200 | | ImageNet | |
|---|---|---|---|---|---|---|---|---|
| | Near-OOD | Far-OOD | Near-OOD | Far-OOD | Near-OOD | Far-OOD | Near-OOD | Far-OOD |
| Synthetic OOD | 91.16 | 93.8 | 78.9 | 84.44 | 83.4 | 92.19 | 78.91 | 94.8 |
| ID-Corrupted Val | 90.55 | 93.88 | 80.27 | 81.41 | 83.9 | 92.58 | 83.5 | 94.6 |
| OpenOOD Val | 91.01 | 94.18 | 80.92 | 80.7 | 81.86 | 94.77 | 78.05 | 96.16 |
| Uniform | 90.9 | 94.1 | 77.26 | 84.5 | 82.84 | 94.46 | 75.23 | 94.55 |
| Mean of Prototypes | 88.5 | 91.4 | 77.2 | 81.4 | 82.3 | 92.9 | 71.25 | 82.21 |

**ID-Corrupted Val** We further validated our approach using 100 i.i.d. samples from CIFAR-10-C Hendrycks & Dietterich (2019) for CIFAR-10 as ID and CIFAR-100C for rest of ID datasets including CIFAR-100, ImageNet-200 and ImageNet to ensure no possible overlap or leaks to the test set.

**Uniform** We try to approximate the representation of OOD data by leveraging uniform noise data. Initially, a batch of random noise images is created using uniformly distributed pixel values across all channels. These noise images are then passed through a neural network to extract logits and features from intermediate layers. An energy score is computed for each image, where lower scores indicate a higher likelihood of being OOD. The images with the lowest energy scores, which are most similar to out-of-distribution (OOD) data, are selected, and their penultimate layer features are extracted. These features are then aggregated to form an OOD prototype.

**Synthetic OOD** To simulate OOD data representations, we utilize a manifold mixup technique on the early layer, similar to the targeted mixup approach described in § 5. However, our method differs in the interpolation target. Instead of interpolating towards the predicted class prototype, we interpolate towards the second-highest predicted class $c_2$, which is the closest incorrect class on the decision boundary.

**Mean of Prototypes** Instead of trying to approximate the representation of OOD data using auxiliary OOD data we rely on the mean of ID prototypes.

Table 6 illustrates our method's robustness to different validation OOD datasets.

### A.3 Density Plots

To comprehensively evaluate the GROOD method's ability to distinguish between in-distribution (ID) and out-of-distribution (OOD) data, we visualize the distribution of OOD scores across a range of datasets with varying characteristics. Figure 4 presents these visualizations for ID, Near-OOD, and Far-OOD samples on CIFAR-10, CIFAR-100 (datasets with relatively small, natural images), ImageNet-200 (a subset of ImageNet), and ImageNet-1k (a large-scale, complex dataset). Analyzing performance across this spectrum demonstrates GROOD's robustness to differences in image complexity and dataset size. In each subplot, we use density plots to represent the distribution of OOD scores, allowing for a clear visual comparison of separation between ID and OOD distributions.

### A.4 Inference Speed

GROOD introduces a novel Out-of-Distribution (OOD) detection method involving two forward passes for the mixup part which is inexpensive to compute, a backward pass over the OOD prototype which can be computed using its closed form expression as in eq. (5) of the main paper and the ne arest neighbor search which is more computationally intensive. For the latter, Our approach employs the FAISS IndexIVF method for efficient distance computation, utilizing centroids and inverted lists instead of the complete dataset. This technique notably enhances inference speed compared to KNN, particularly in our CIFAR benchmarks. Specifically, on CIFAR-10 and CIFAR-100 datasets, GROOD recorded evaluation inference times over all OOD test sets of **130** seconds and **155** seconds, respectively. This is significantly faster than KNN, which took 434 seconds for CIFAR-10 and 641 seconds for CIFAR-100, demonstrating the efficiency of our approach.

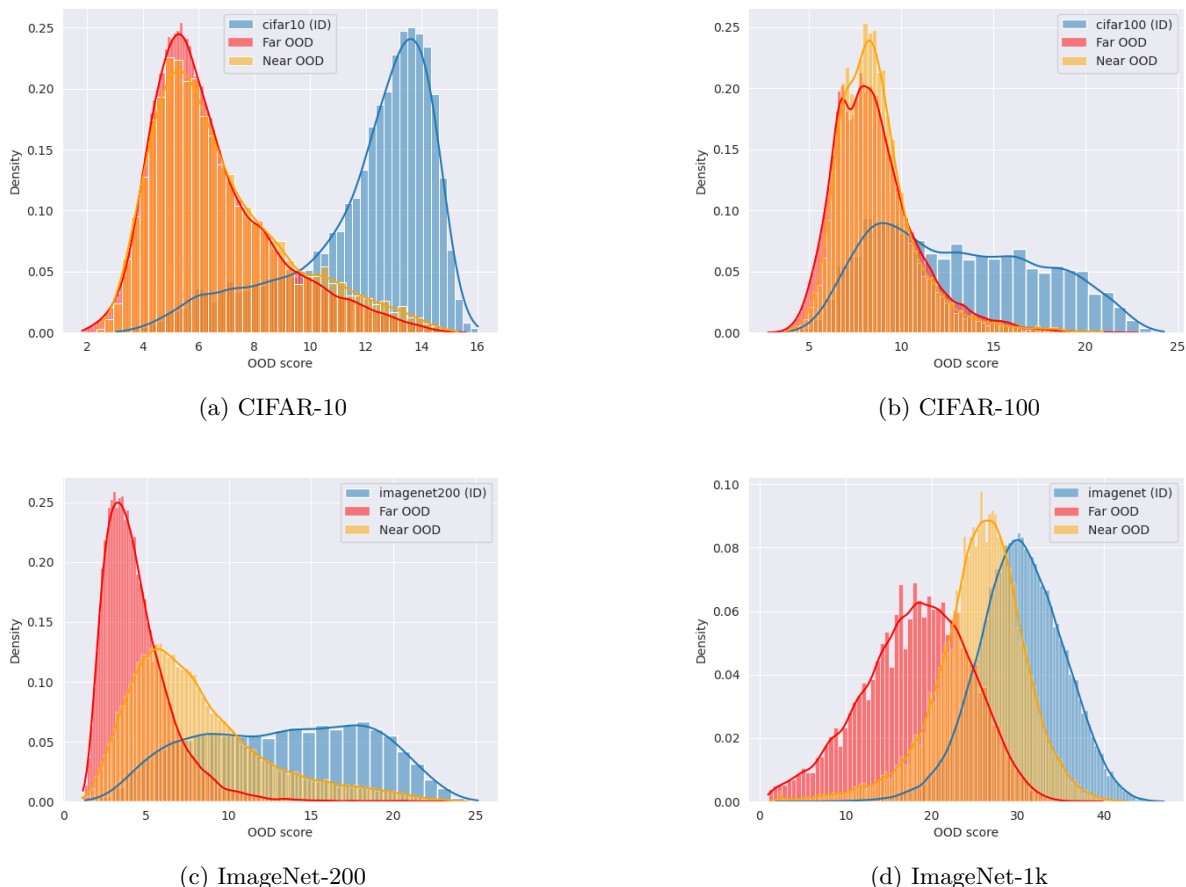

Figure 4: Distribution of OOD scores on Near-OOD and Far-OOD across different datasets.

### A.5 Ablate value of kth nearest neighbor

Figure 5 illustrates the impact of using the distance to the $k$-th nearest neighbor, as proposed by Sun et al. (2022). The plot demonstrates that employing the distance to the nearest point in gradient space leads to optimal results.

### A.6 Gradient computation in closed form

This section details the derivation of the closed-form expression for the gradient, as presented in eq. (5) of the main paper. The cross-entropy loss for a logit $L = (L_j)_{j=1}^C + 1$ and some (any) ID label $y$ is given by:

$$H(L, y) = -\ln \frac{\exp(L_y)}{\sum_{i=1}^{C+1} \exp(L_j)}$$

The partial derivative of this loss with respect to $L_{C+1}$ is given by:

$$\frac{\partial}{\partial L_{C+1}} H(L, C+1) = \frac{\exp(L_{C+1})}{\sum_{i=1}^{C+1} \exp(L_j)}$$

which is equal to the softmax probability corresponding to the OOD class and does not depend on the specific ID class label $y$ anymore. Now for a feature vector $h$, the corresponding logit vector $L(h)$ is given by eq. (2). Since $L_{C+1}(h) = \|h - \mathbf{p}_{\text{ood}}^{\text{pen}}\|_2$ is the only logit depending on $\mathbf{p}_{\text{ood}}^{\text{pen}}$, the gradient of the above loss

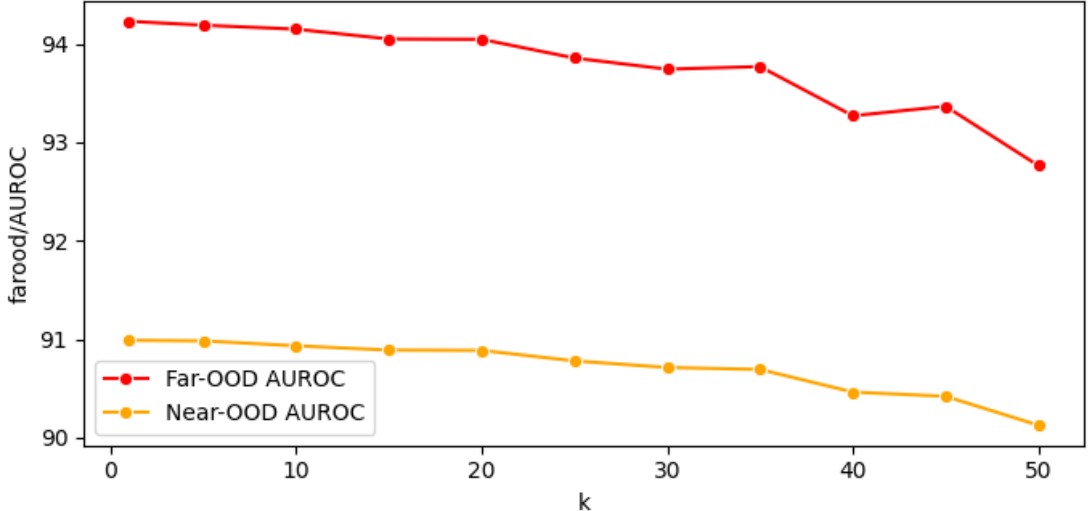

Figure 5: Ablating different values for k-th nearest neighbor parameter on CIFAR-10

with respect to $\mathbf{p}_{\text{ood}}^{\text{pen}}$ is given by the chain rule:

$$\nabla_{\mathbf{p}_{\text{ood}}^{\text{pen}}} H(L(h), y) = \frac{\partial}{\partial L} H(L(h), y) \nabla_{\mathbf{p}_{\text{ood}}^{\text{pen}}} L(h) = \frac{\exp(L_{C+1})}{\sum_{i=1}^{C+1} \exp(L_j)} \frac{h - \mathbf{p}_{\text{ood}}^{\text{pen}}}{\|h - \mathbf{p}_{\text{ood}}^{\text{pen}}\|_2} = p_{\text{ood}}(h) \frac{h - \mathbf{p}_{\text{ood}}^{\text{pen}}}{\|h - \mathbf{p}_{\text{ood}}^{\text{pen}}\|_2},$$

as desired.

### A.7 Impact of Mixup-Trained Backbones on GROOD

We investigate how GROOD's OOD prototype construction interacts with backbones trained using manifold mixup (Verma et al., 2019). Since these models are explicitly trained to generalize across interpolated samples, we hypothesized that synthetic mixup-based OOD prototypes might no longer serve as an effective deviation reference.

| Method | Near-OOD AUROC (%) | Far-OOD AUROC (%) |
|---|---|---|
| GROOD (mean prototype) | **81.05** | **80.26** |
| ASH (Djurisic et al., 2022) | 79.1 | 56.0 |
| KNN (Sun et al., 2022) | 78.0 | **81.85** |
| GradNorm (Huang et al., 2021b) | 50.0 | 50.0 |

Table 7: Performance of GROOD with mean prototype on a manifold mixup-trained ResNet-18 backbone (CIFAR-100).

To test this, we evaluated GROOD on a ResNet-18 trained with manifold mixup. Instead of using mixup-based OOD prototypes since it will no longer represent OOD data, we used a mean prototype computed from ID class prototypes. The results are shown below:

These results show that GROOD maintains competitive performance by adjusting its prototype strategy to the model's training procedure. The mixup-based prototype remains optimal for standard-trained models, while alternative strategies like mean prototypes are preferable when the backbone is mixup-regularized.

