# OpenReview forum: "GROOD: GRadient-Aware Out-of-Distribution Detection"
_TMLR — Accepted by TMLR_

### Review · Reviewer_zBtZ · 2025-05-13

**Summary Of Contributions:**

The paper introduces GROOD, a novel post-hoc method for out-of-distribution (OOD) detection that leverages an artificial OOD prototype in the feature space of a pre-trained deep neural network. By analyzing the gradients of a nearest-class-prototype loss with respect to this synthetic OOD prototype—instead of traditional gradients with respect to model parameters—GROOD effectively distinguishes between in-distribution (ID) and OOD samples, as OOD data exhibits greater sensitivity to changes in the OOD prototype. The method integrates insights from neural collapse, utilizes prototype geometry, and introduces a mixup-based approach for generating synthetic OOD samples, requiring no extra training or model modifications. Extensive experiments show that GROOD achieves superior OOD detection performance and robustness, especially for challenging datasets like ImageNet-1k, highlighting the benefits of combining feature and gradient space analyses in OOD detection.

**Audience:**

Yes

**Claims And Evidence:**

Yes

**Requested Changes:**

1. Traditionally, neural collapse (NC) is often considered detrimental for OOD detection because it leads to overconfidence. However, this paper presents an interesting application of neural collapse. I believe this aspect deserves more discussion and should be highlighted in the paper.

2. The proposed method is closely related to energy-based OOD detection methods. For example, equation (4) is essentially an energy function, and its gradient serves as a score function. Therefore, the connections with existing energy-based methods should be discussed, as well as whether the techniques proposed in this paper can be applied to other methods.

3. The method is also related to feature-based approaches, particularly those using Mahalanobis distance and cosine similarity from class centroids. A discussion of these relationships is needed. Furthermore, can these distances replace the L2 distance in equation (2), and how would their performance compare?

4. This paper employs synthetic OOD generation using mixup. However, this approach likely covers only a small subset of possible OOD samples and may lead to inaccurate estimation of the OOD prototype. How significant is the gap between synthetic OOD samples and the global oracle?

5. An interesting observation from Figure 2(a) is that the OOD points are “inside” the ID points. Why does this occur?

**Strengths And Weaknesses:**

**Strengths:**

1. The proposed method is simple yet effective, achieving state-of-the-art performance on various benchmarks, especially on near-OOD tasks.

2. It introduces an interesting application of neural collapse.

3. The related work section is extensive and accessible to a general audience.

4. The proposed method demonstrates deep connections to all existing paradigms.

Overall, this paper is solid and does not have many weaknesses. However, there are a few areas for improvement:

1. The paper appears to have deeper connections with existing paradigms, which should be discussed further to increase its impact. (Changes 1-3)

2. More discussion of the design choices and experiments is needed. (Changes 4,5)

---

> ### Author Response · Authors · 2025-06-23
>
> ## Concern 1: The Role of Neural Collapse
>
> Thank you for your insightful comment. We agree that our use of Neural Collapse (NC) in GROOD warrants more emphasis. While NC is often seen as detrimental for OOD detection due to overconfident predictions, GROOD leverages it differently.
>
> NC induces a highly structured ID feature space where samples cluster tightly around class prototypes, making deviations more identifiable. Instead of relying on raw confidence, GROOD introduces an artificial OOD prototype and analyzes the gradients of a prototype-based loss with respect to it. ID samples exhibit low sensitivity and stable gradients, while OOD samples show high sensitivity or directional misalignment. This transforms NC into a useful asset for OOD detection—a point we will highlight more clearly in the Introduction and Section 4.1.
>
> ## Concern 2: Connection to Energy-Based Methods
>
> We appreciate the connection to energy-based OOD methods. While Equation (4), derived from prototype distances, can be interpreted as an energy function, GROOD diverges from standard energy-based detection.
>
> Rather than using the energy value directly, GROOD’s novelty lies in analyzing the gradient of the energy (loss) with respect to the OOD prototype (Eq. 5). This gradient captures how sensitively a sample reacts to the prototype. We then compute the final OOD score using nearest-neighbor distances in this gradient space (Eq. 6), distinguishing GROOD from conventional energy-based scoring approaches.
>
> ## Concern 3: Connection to Feature-Based Methods
>
> We acknowledge that GROOD operates in the feature space and utilizes class prototypes (centroids), similar to methods employing Mahalanobis distance or cosine similarity to measure distance to learned feature representations. However, GROOD distinguishes itself by moving beyond direct feature-space distance measurements for its final OOD score.
>
> Instead, our novelty lies in analyzing the gradients of a prototype-based loss with respect to an artificial OOD prototype, and then performing nearest-neighbor detection in this gradient space. This approach provides a unique perspective, capturing subtle differences between ID and OOD samples that might not be apparent through direct feature similarity metrics alone.
>
> While, in principle, other distance metrics could replace L2 in Equation (2), our choice of L2 distance is fundamental to GROOD's core mechanism and its derived properties. As explained in our previous response, using L2 distance directly enables the elegant and interpretable closed-form gradient expression in Equation (5). This specific gradient, where the norm represents the OOD probability and the direction indicates the deviation from the OOD prototype, is crucial for GROOD's ability to capture both magnitude and direction of out-of-distribution likelihood. Alternative metrics like cosine similarity would yield different and potentially less interpretable gradient expressions, compromising this key aspect of our design.
>
> ## Concern 4: Limitations of Synthetic OOD Generation
>
> The "Global Oracle" experiment (Table 1)provides insight into the effectiveness  our approach when we have access to a small portion of actual diverse OOD data from all other OOD datasets to construct a single general OOD prototype.
>  Our synthetic OOD prototype, in contrast, is derived solely from in-distribution samples through a strategic mixup process near decision boundaries, designed for practical scenarios where no auxiliary OOD data is available.
> Comparing the AUROC performance between GROOD utilizing a synthetic OOD prototype (Table 6) and the Global Oracle (Table 1), we observe the following gaps (Synthetic OOD AUROC - Global Oracle AUROC):
>
> - CIFAR-10: Near-OOD: +1.16% (91.16% vs 90.8%) , Far-OOD: −1.0% (93.8% vs 94.8%)
> - CIFAR-100: Near-OOD: −3.66% (84.44% vs 88.1%) , Far-OOD: −4% (78.9% vs 82.1%)
> - ImageNet-200: Near-OOD: -0.7% (83.4% vs 84.1%) , Far-OOD: -2.03% (92.19% vs 94.22%)
> - ImageNet-1K: Near-OOD: −0.09% (78.91% vs 79%) , Far-OOD: −1.4% (94.8% vs 96.2%)
>
> As expected, a gap exists, particularly notable for ImageNet-1K Far-OOD and CIFAR-100 Far-OOD, where the Oracle performs significantly better. This highlights the potential gains achievable with access to diverse, real OOD data.
>
> ## Concern 5: OOD Points Appearing "Inside" ID Points
>
> This behavior occurs because networks are trained solely on ID data, mapping OOD samples into their learned feature space without a specific "unknown" region. This intermingling is precisely the challenge GROOD addresses; by analyzing gradients in relation to an OOD prototype, we achieve clearer separation in the gradient space, as shown in Figure 2(b)

---

### Review · Reviewer_ZBKA · 2025-05-26

**Summary Of Contributions:**

This paper proposes Gradient-Aware Out-of-Distribution Detection (GROOD), a new OOD detection method inspired by neural collapse theory and prototype-based classification. The authors introduce an out-of-distribution prototype that serves as a fixed point for measuring the OOD degree of new samples. Experimental results demonstrate that GROOD outperforms existing methods in AUROC across a wide range of datasets.

**Audience:**

Yes

**Claims And Evidence:**

Yes

**Requested Changes:**

- Please see weakness for a range of points that require clarification and improvement.

- It seems like all the methods compared are published before 2023. It would strengthen the evaluation to include more recent methods or benchmarks.

- While the availability of open-source code is appreciated, it would be helpful to include a clear textual description of the training and evaluation setup, such as number of epochs, learning rate schedule, and batch size, in the main paper or appendix.

- Page 9, please fix the incorrect citation format of Yang et al. (2022).

**Strengths And Weaknesses:**

**Strengths:**

- The idea is straightforward and easy to understand. The author provided sufficient context (albeit not well organized, see below) for readers.

- Experiments are conducted on a wide range of datasets that follows the setting of previous works.


**Weaknesses:**

- Several design choices are not sufficiently explained or justified.

  - In the intro (P2), the authors present two key observations that motivate GROOD. While these observations are relevant, their presentation feels abrupt and disconnected from the surrounding text. It may improve clarity to briefly summarize these insights in the Introduction and elaborate on them in the Methods section (Section 4.1). Rewriting this part to enhance logical flow would greatly improve readability.

  - In Eq2, authors use the L2 distance between a sample and each prototype as the logit vector. This design choice is not clearly justified. Since Euclidean distance may not always reflect meaningful similarity in embedding space, alternative metrics like cosine similarity may be more appropriate. Clarification/experiments on why L2 distance was chosen is recommended.

  - The proposed method relies on the construction of a single OOD prototype, as described in Figure 2 and Section 5.2. However, it is unclear whether OOD samples naturally form a coherent group. The assumption that a single prototype can represent diverse OOD samples may not hold, and the explanation in Section 5.2 does not adequately address why multiple OOD prototypes were not considered. A more detailed justification of this design would strengthen the paper.

---

> ### Author Response · Authors · 2025-06-23
>
> ## Concern 1: Logical Flow of Introduction
>
> We appreciate your constructive suggestion to briefly summarize these insights in the Introduction and elaborate on them more thoroughly in the Methods section (Section 4.1), where their operational role in GROOD's mechanics is described.
> We will revise the manuscript to implement this suggestion. Specifically:
>
> - In the Introduction, we will refine the summary of these two key observations to concisely state their core motivation for GROOD, ensuring a smoother transition and better context for the reader.
>
> - The detailed elaboration on how the Neural Collapse property motivates prototype-based classification and how the dispersed nature of OOD samples informs the strategic use of an artificial OOD prototype and gradient analysis will be further emphasized and thoroughly explained within Section 4.1, Gradients Computation.
>
> ## Concern 2: Justification for Using L2 Distance
>
> Our choice of L2 distance is not arbitrary; it is meticulously selected to ensure the core mechanism of GROOD functions effectively and robustly. The primary reason is that using L2 distance allows the gradient of the loss function with respect to the OOD prototype (∇H(h)) to simplify into the elegant and highly interpretable **closed-form expression presented in Equation (5)**. This expression reveals that the norm of the gradient is precisely the softmax probability of the sample being OOD (pood​(h)), and its direction is given by the unit vector pointing from the OOD prototype to the sample's feature (h−poodpen​)/∣∣h−poodpen​∣∣2​). This direct coupling of gradient magnitude to OOD likelihood and gradient direction to feature deviation is fundamental to GROOD's ability to measure both direction and magnitude of deviation, which would not be the case if alternative metrics like cosine similarity were used for gradient computation.
> For instance, on the CIFAR-100 dataset using cosine similarity resulted in **70.13% Near-OOD AUROC** and **84.73% Far-OOD AUROC**. These results quantitatively demonstrate that L2 distance provides superior discrimination compared to cosine similarity within our framework.
>
> Beyond the gradient properties, the L2 distance also aligns perfectly with the geometric insights provided by **the Neural Collapse (NC)** property. NC posits that features of in-distribution data collapse to their class means in the feature space, forming compact, well-separated clusters. In such a Euclidean space, L2 distance naturally serves as an effective metric for measuring proximity to these mean-based prototypes.
>
> ## Concern 3: Justification for a Single OOD Prototype
>
> See reviewer 1 concern 2
>
> ## Concern 4: Comparison with More Recent Baselines
>
> We will try adding more recent baselines including fdbd and NCI.
>
> ## Concern 5: Missing Experimental Setup Details
>
> We used the checkpoints available in torchvision following their standard training strategy,  we will add this along with a citation including how the training was done.
>
> ## Concern 6: Citation Format
>
> Thank you for bringing to our attention the incorrect citation format for Yang et al. (2022) on Page 9. We apologize for this oversight.
> We will promptly correct the citation format in the revised manuscript.

---

> > ### Comment · Reviewer_ZBKA · 2025-06-27
> >
> > Thank you for your revision. While the answers regarding all other concerns are addressed, I am still not fully convinced over the justification for a single OOD prototype. Would you be able to elaborate on how the gradient vectors would become outliers? The formatting of the official comment for reviewer 7fnF is also incorrect.

---

> > > ### Author Response · Authors · 2025-06-30
> > >
> > > Thank you for your follow-up. We appreciate your close attention to the conceptual foundations of our method. Regarding your question:
> > >
> > > We would like to clarify that the single OOD prototype in GROOD does **not** attempt to geometrically represent the full diversity of OOD space. Instead, it serves as a fixed reference point for computing the **sensitivity** of a sample’s feature representation to a hypothetical OOD perturbation.
> > >
> > > This sensitivity is captured by the gradient of the NCP softmax loss with respect to the OOD prototype. As shown in Eq. (5), this gradient is proportional to the softmax probability of the OOD class and points in the direction from the OOD prototype to the feature vector. For ID samples, the loss is minimally sensitive to the OOD prototype, leading to small, directionally coherent gradients. In contrast, OOD samples lying outside the structure imposed by neural collapse exhibit greater response variability and misalignment, leading to gradient vectors that **deviate significantly** in both magnitude and direction.
> > >
> > > As illustrated in Fig. 2(b), this causes OOD gradients to appear as **outliers in gradient space**, where we apply nearest-neighbor scoring. This transformation consistently improves ID/OOD separability, even in cases where feature-space separation is ambiguous.
> > >
> > > Thus, the role of the OOD prototype is not to cover OOD diversity, but to induce a discriminative **gradient response manifold**. This is supported by both the improved AUROC metrics (Table 2) and oracle experiments (§5.1) showing that even a single, imperfect prototype suffices to generate meaningful contrast.
> > >
> > > We hope this clarifies the rationale and utility behind our use of a single OOD prototype in GROOD.
> > >
> > > Finally, thank you for pointing out the formatting issue. We will revise the official comment for reviewer 7fnF accordingly.

---

### Review · Reviewer_7fnF · 2025-06-12

**Summary Of Contributions:**

The paper proposes GROOD, a post-hoc OOD detection method leveraging the gradient of a nearest-class-prototype (NCP) loss with respect to an OOD prototype. GROOD is proposed based on the intuition that gradients derived from ID and OOD samples differ significantly in magnitude and direction, enabling better separation in the gradient space. It introduces synthetic OOD prototype generation via mixup and includes a thorough empirical analysis. GROOD avoids retraining, achieves high performance across several benchmarks and is demonstrated to generalize across architectures.

**Audience:**

Yes

**Broader Impact Concerns:**

Not applicable.

**Claims And Evidence:**

Yes

**Requested Changes:**

1. The current performance on CIFAR-100 and ImageNet-1k appears less competitive compared to several recent OpenOOD baselines. Please provide analysis to explain why GROOD may underperform on these datasets.

2. Please provide a more detailed explanation or theoretical justification for why a single OOD prototype is sufficient to capture the diversity of real-world OOD samples, especially when GROOD assumes this for all test-time OOD detection. It would be more convincing if the OOD test data has multiple modes such as a Gaussian mixture model.

3. The current presentation lacks in-depth analysis on what specific challenge GROOD is designed to solve compared to existing post-hoc or gradient-based OOD methods.

4. Discuss the Role of Mixup training and Its Impact on test-time GROOD

**Strengths And Weaknesses:**

**Strengths**

The paper is clearly written and the proposed method is clearly presented and easy to follow.

The experimental settings are standard and the results show the strength of GROOD over a few baselines.


**Weaknesses**

It seems that the performance of the method is not competitive compared with recent baselines particularly on CIFAR100 and ImageNet-1k, according to OpenOOD benchmark https://zjysteven.github.io/OpenOOD/.

The paper claims that “While real-world OOD samples exhibit considerable diversity, representing them with a single prototype proves effective within our gradient-aware framework.” It looks like the gradient distance is a unit vector with the length equal to confidence score. I don’t get why this vector can handle the diversity of OOD samples. The experiment looks good but the paper does not seem to explain what challenge the proposed method is handling and why quite well.

The GROOD method seems to favor the mixup-based prototype, but it is unknow how mixup training will affect the effectiveness of GROOD.

---

> ### Author Response · Authors · 2025-06-23
>
> ## Concern 1: Performance on CIFAR-100 and ImageNet-1k
>
> We acknowledge the OOD landscape's trade-offs. No single method dominates all benchmarks, but GROOD excels in specific areas.
>
> * **CIFAR-100**: GROOD achieves **SOTA Far-OOD AUROC (84.44%)** among compared methods (Table 2). Its **Near-OOD (78.9%)** is highly competitive with top post-hoc methods like VIM (82.40%), surpassing others like Wei et al. + KLD (79.01% Far, 81.37% Near) and NAC (86.9% Far, 75.9% Near). We note that higher leaderboard scores often come from non-post-hoc methods requiring outlier training.
> * **ImageNet-1k**: GROOD achieves a superior **Far-OOD score (94.8%)**, while methods like CombOOD prioritize Near-OOD.
>
> GROOD's performance links to the ID feature space quality and OOD prototype selection. Our Neural Collapse-inspired approach relies on well-clustered ID representations. Prototype choice manages the Near-/Far-OOD trade-off: a synthetic prototype prioritizes **Far-OOD (84.44% on CIFAR-100)**, while an "ID-Corrupted Val" prototype offers **balanced performance (80.27% Near, 81.41% Far on CIFAR-100; 83.5% Near, 94.6% Far on ImageNet-1K)**. This highlights GROOD's flexibility. We will add this context to Section 6.
>
> ## Concern 2: Justification for a Single OOD Prototype
>
> The single OOD prototype acts as a **strategic gradient-based reference point**, not a geometric representation of OOD space. its efficacy comes from serving as a consistent reference point for gradient computations, which allows us to effectively detect deviations.
>
> As per Equation (5), the gradient has its norm equal to $$p_{ood}(h)$$ and its direction points from $$p_{ood}^{pen}$$ to h.
> * **ID samples:** Confidently classified samples, $$p_{ood}(h)$$ is low, leading to small, consistently directed gradients. Their OOD scores remain low.
> * **OOD samples:** These don't align with learned ID manifolds. Their gradient vectors are outliers in **magnitude** (if $$h$$ is near $$p_{ood}^{pen}$$, causing high $$p_{ood}(h)$$) or **direction** (if $$h$$ is far from all prototypes). This enables detection without geometric proximity to $$p_{ood}^{pen}$$.
>
> Thus, $$p_{ood}^{pen}$$ serves as a fixed reference. The gradient's magnitude and direction provide a nuanced signal for ID/OOD distinction, as visually shown in Figure 2(b). We will incorporate this detailed explanation.
>
> ## Concern 3: In-depth Analysis of Challenges Addressed by GROOD
>
> GROOD addresses key challenges in post-hoc OOD detection:
> * **Near-OOD Samples:** Achieves effective distinction using unique gradient sensitivity patterns from an artificial OOD prototype and class prototypes a task where many current approaches struggle.
> * **Robustness:** Offers enhanced performance across diverse architectures, including Transformers, where many methods degrade A AHOQN IN Table 3.
> * **Simplicity:** Significantly reduces hyper-parameter tuning needs.
> * **Stability:** Contributes to more stable OOD detection performance throughout training checkpoints as shown in Table 5.
>
> ## Concern 4: Role and Impact of Mixup Training
>
> We distinguish mixup as a **backbone training strategy** from **mixup for crafting GROOD's OOD prototype**.
>
> GROOD is post-hoc. Our primary OOD prototype construction uses mixup-based synthetic data, highly effective for models *not* trained with mixup regularization, generating strong references for gradient sensitivity near decision boundaries.
>
> However, if the **backbone model is trained with manifold mixup**, using mixup-based synthetic data for the OOD prototype becomes less effective, as the model may interpret it as in-distribution. In such cases, alternative OOD prototype constructions are crucial. For a manifold mixup-trained checkpoint, a mean of ID prototypes prototype yielded **81.05% AUROC for Near-OOD** and **80.26% AUROC for Far-OOD**.
>
> * GROOD (81.05%) outperforms the best Near-OOD baseline (ASH, 79.1%).
> * For Far-OOD, GROOD (80.26%) is highly competitive with the best baseline (KNN, 81.85% Far, 78% Near).
>
> This demonstrates GROOD's flexibility; while our main results use mixup-generated OOD prototypes for standard models, the method is robust enough to employ alternative prototypes when the base model's training regimen requires it. We will clarify this in the revised manuscript's supplementary area Table 7.

---

### Decision · Action_Editor_TD35 · 2025-08-14

**Recommendation:** Accept with minor revision

**Additional Comments:**

There are some additional efforts for the authours, including,
1) Provide a clearer, earlier explanation in the introduction of the two key motivating observations, linking them smoothly to the proposed method.
2) Strengthen justification for using a single OOD prototype, possibly with more visual or empirical evidence for its sufficiency.
Include more recent baselines in the comparisons to position the contribution within the latest state of the art.
3) Add concise training/evaluation setup details (epochs, learning rate schedule, batch size) in the main text or appendix.
Improve discussion of design choices (e.g., why L2 distance is preferred) and potential limitations, such as scenarios where mixup-based prototypes may underperform.

**Audience:**

Yes

**Audience Explanation:**

Yes. The topic of out-of-distribution (OOD) detection—especially post-hoc methods leveraging gradient analysis—remains relevant to the TMLR readership, particularly those interested in model reliability, robustness, and interpretability.

**Claims And Evidence:**

Yes

**Claims Explanation:**

Yes. The reviewers generally agreed that the claims are supported by thorough experiments across multiple datasets, solid theoretical grounding, and clear explanations. The authors provided detailed rebuttals addressing performance trade-offs, the rationale for a single OOD prototype, and the role of mixup-generated prototypes, which strengthened the paper’s validity.